# Activating cobalt(II) oxide nanorods for efficient electrocatalysis by strain engineering

Tao Ling[1,2], Dong-Yang Yan[1], Hui Wang[3], Yan Jiao[2], Zhenpeng Hu [4], Yao Zheng[2], Lirong Zheng[5], Jing Mao[1], Hui Liu[1], Xi-Wen Du[1], Mietek Jaroniec [6] & Shi-Zhang Qiao[1,2]

Designing high-performance and cost-effective electrocatalysts toward oxygen evolution and hydrogen evolution reactions in water–alkali electrolyzers is pivotal for large-scale and sustainable hydrogen production. Earth-abundant transition metal oxide-based catalysts are particularly active for oxygen evolution reaction; however, they are generally considered inactive toward hydrogen evolution reaction. Here, we show that strain engineering of the outermost surface of cobalt(II) oxide nanorods can turn them into efficient electrocatalysts for the hydrogen evolution reaction. They are competitive with the best electrocatalysts for this reaction in alkaline media so far. Our theoretical and experimental results demonstrate that the tensile strain strongly couples the atomic, electronic structure properties and the activity of the cobalt(II) oxide surface, which results in the creation of a large quantity of oxygen vacancies that facilitate water dissociation, and fine tunes the electronic structure to weaken hydrogen adsorption toward the optimum region.

[1] Key Laboratory for Advanced Ceramics and Machining Technology of Ministry of Education, Institute of New-Energy, School of Materials Science and Engineering, Tianjin University, Tianjin 300072, China. [2] School of Chemical Engineering, The University of Adelaide, Adelaide, SA 5005, Australia. [3] Key Laboratory of Aerospace Materials and Performance (Ministry of Education), School of Materials Science and Engineering, Beihang University, Beijing 100191, China. [4] School of Physics, Nankai University, Tianjin 300071, China. [5] Beijing Synchrotron Radiation Facility, Institute of High Energy Physics, Chinese Academy of Sciences, Beijing 100049, China. [6] Department of Chemistry and Biochemistry, Kent State University, Kent, OH 44242, USA. Tao Ling, Dong-Yang Yan, Hui Wang and Yan Jiao contributed equally to this work. Correspondence and requests for materials should be addressed to S.-Z.Q. (email: s.qiao@adelaide.edu.au)

The hydrogen evolution reaction (HER), a half-reaction of water splitting, plays a key role in many sustainable energy conversion technologies, such as electrolysis[1,2], photo-electrochemical water splitting[3,4], etc. Currently, platinum (Pt)-based catalysts are still the most efficient and durable HER catalysts in both acid and alkaline media[5–8]. However, the widespread application of the aforementioned technologies is hampered by the challenge in developing high-performance yet cost-effective catalysts[9,10]. Due to the availability of cost-effective oxygen-evolution reaction catalysts (OER—another reaction of water splitting) on the counter electrode in alkaline media[4,11,12], tremendous efforts have been undertaken toward the development of highly efficient and durable HER catalysts in alkaline solutions to achieve sustainable hydrogen production.

During the past decade, transition metal oxides (TMOs) have emerged as particularly promising nonprecious metal-based OER catalysts in alkaline environment[13–20]. Recently, TMOs were coupled with (noble) metals to facilitate HER under alkaline conditions[5,21–25]. However, pure TMOs are generally considered as HER-inactive materials due to their inappropriate hydrogen adsorption energy[10,23]. So far, a fundamental understanding of HER mechanism on TMOs is still lacking, and whether a pure TMO surface can exhibit a comparable catalytic activity and be even more active than Pt-based electrocatalysts is still unknown.

One common strategy to enhance the activity of electrocatalysts is to tune their surface electronic structure[26]. Strain engineering, expanded, or compressed arrangements of atoms, are one of the promising routes to manipulate the surface electronic structure of electrocatalysts[27–34]. Currently, the most electroactive Pt-based catalysts were achieved by adopting transition metal atoms in the underlying atomic layers through the compressive strain on the Pt surface and in turn, showed an improved electrocatalytic activity[27,32,35–37]. In the case of TMOs, strain effect was introduced to modulate the electronic structure of TMOs, e.g., iron oxides[38–40]. Moreover, theoretical and experimental studies demonstrated that the planar strain between the epitaxial thin TMO films and lattice-mismatched substrates can alter the surface chemistry of TMO by impacting the surface oxygen stoichiometry[41,42]. These strained TMO films are of crucial importance to fundamental research, but their practical applications have been limited by insufficient catalytic activity and durability. Hence, the incorporation of lattice strain in high-surface-area TMO nanostructures and the study of strong strain coupling of the atomic, electronic structure, and catalytic properties of these nanostructures are of broad scientific significance and great technological importance.

As an emerging catalytic material for OER[43] and photocatalytic hydrogen evolution[44], cobalt(II) oxide (CoO) has received considerable attention. Herein, we report an enhancement of the alkaline HER activity of CoO nanorods (NRs) fabricated by surface strain engineering. Our theoretical and experimental results suggest that the tensile strain can strongly couple the atomic, electronic structures and HER activity of CoO NRs. This generates abundant oxygen (O) vacancies that promote water dissociation and weaken the hydrogen adsorption toward the optimum region. As a result, the surface-strained CoO NRs exhibit high intrinsic HER activity, assessed on the basis of both hydrogen adsorption free energy ($\Delta G_{H^*}$) and exchange current density ($j_0$), which surpasses all the previously reported efficient HER catalysts in alkaline solutions, including the state-of-the-art Pt/C catalysts, to the best of our knowledge. This strain-tuneable TMO opens up opportunities for the design of superior electrocatalysts.

## Results

**Alkaline hydrogen evolution mechanism on the surface of CoO.** HER is acknowledged to proceed via either Volmer–Heyrovsky or Volmer–Tafel pathways in alkaline media[45] (Volmer: $* + H_2O + e^- \rightleftharpoons H^* + OH^-$; Heyrovsky: $H^* + H_2O + e^- \rightleftharpoons H_2 + OH^-$; and Tafel: $2H^* \rightleftharpoons H_2$, where $*$ is the active site). Although TMOs are generally considered to be inappropriate catalysts for converting $H^*$ to $H_2$ (excessively strong $H^*$ adsorption on O, but extremely weak $H^*$ adsorption on metal ion)[5,10,23], they have been shown to be very active for the OH–H bond cleavage during the Volmer step[5,21,24,46]. In particular, we investigated water dissociation on CoO based on the spin-polarized density functional theory plus U (DFT+U) calculations (Supplementary Figs. 1–3 and Supplementary Note 1). Our results reveal that facile water dissociation can be achieved on the O-vacancy-enriched {111}-O surface of CoO (terminated with O atoms), resulting in OH healing of the O-vacancies and the remaining H atoms adsorbing on the top of the nearest-surface O atoms (Fig. 1a). This is consistent with previous experimental and theoretical reports in that the O-vacancies on the surface of the TMO are the active sites for water dissociation[46].

Afterward, we investigated hydrogen adsorption on the CoO surface. The hydrogen adsorption free energy, $\Delta G_{H^*}$, is a well-known descriptor of the HER activity. The optimum value of $|\Delta G_{H^*}|$ should be zero[47], indicating that hydrogen adsorption is neither too strong nor too weak. Our computational results show that $\Delta G_{H^*}$ on typical CoO {100}, {110}, and {111}-O surfaces is significantly stronger than optimum (Supplementary Figs. 4–6), which is unfavorable for $H^*$ desorption and subsequent $H_2$ production. This means that TMOs are traditionally considered as HER-inactive materials[5,10]. Surprisingly, when tensile strain was exerted on O-vacancy-rich {111}-O surface (hereafter, referred to as "{111}-Ov surface", with 11.1% surface O-vacancies, Supplementary Fig. 7), $H^*$ adsorption was continuously weakened with increasing magnitude of the applied strain (Fig. 1b).

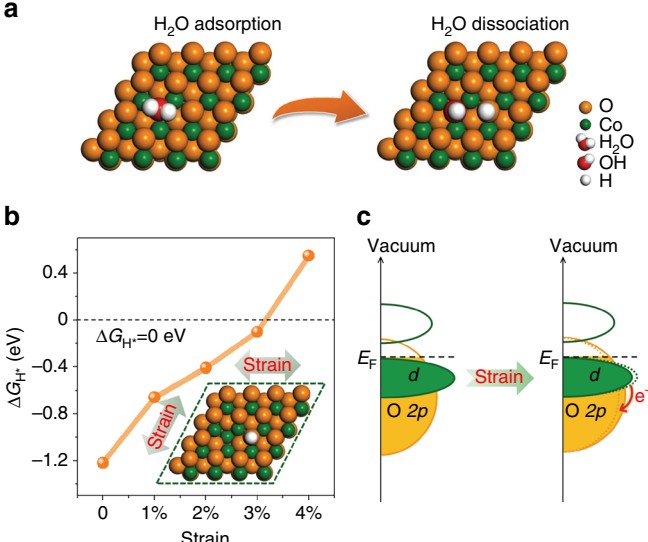

**Fig. 1** Computational predictions for the strain effect on the HER activity of CoO. **a** Schematic illustration of $H_2O$ adsorption and dissociation on the CoO {111} surface with O-vacancies, where $H_2O$ adsorbs onto the site of O-vacancy (left panel), dissociated OH heals the O-vacancy, and the remaining H atom adsorbs on the top of a nearest-surface O atom to form OH state (right panel). **b** Hydrogen adsorption free energy, $\Delta G_{H^*}$, vs. tensile strain for the CoO {111}-Ov surface. The surface O-vacancy concentration on {111}-Ov is ~11.1%. **c** Schematic illustration of the effect of strain on the electronic structure of {111}-Ov surface of CoO

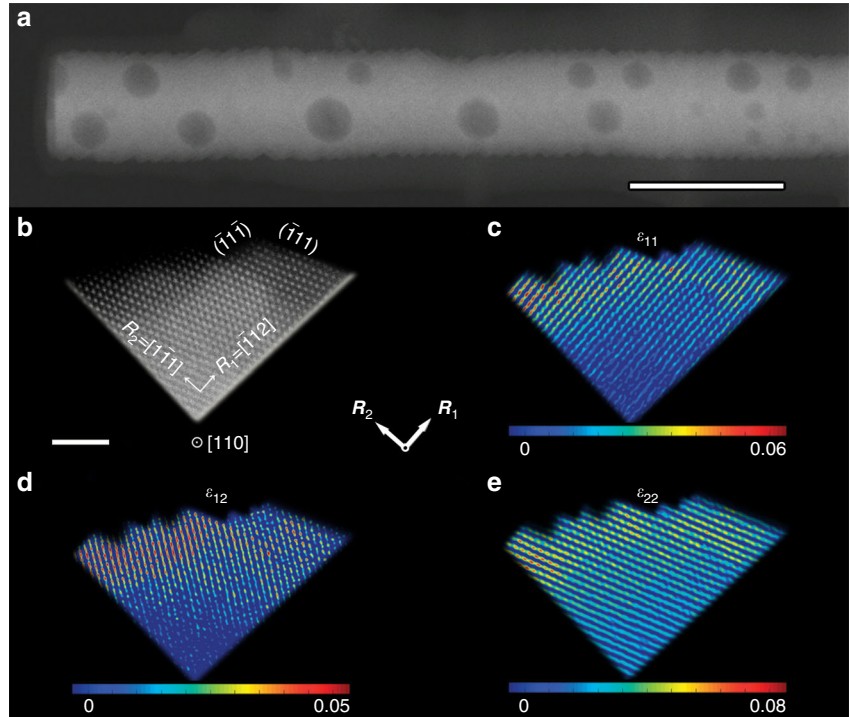

**Fig. 2** Analysis of strain on the surface of S-CoO nanorods. **a** Typical HADDF-STEM image of an individual S-CoO NR, the surface of which is terminated with continuous nano-sawtooths. Scale bar, 100 nm. **b** Atomic resolution HADDF-STEM image of two adjacent nano-sawtooths enclosed with {111} nanofacets, indicating the lattice vectors $R_1$ and $R_2$ used as a reference for the strain analysis. Scale bar, 2 nm. **c-e** Contour plots of the strain component $\varepsilon_{11}$ (**c**), $\varepsilon_{12}$ (**d**), and $\varepsilon_{22}$ (**e**) relative to the reference values

Impressively, 3.0% strain resulted in $\Delta G_{H^*}$ of −0.10 eV, which is close to the optimal value of $\Delta G_{H^*} = 0$ eV.

Next, the role of tensile strain in enhancing the HER activity of the CoO {111}-Ov surface is revealed through the investigation of the electronic structure. As can be seen in Fig. 1b, the adsorption of atomic H on the {111}-Ov surface is assumed to be through coupling of H 1s to the occupied O 2p-state. Band structures of CoO {111}-Ov with 0 and 3% tensile strain are schematically shown in Fig. 1c. As can be seen, the tensile strain upshifts the O 2p-band of CoO (Supplementary Fig. 8), resulting in greater covalency of the Co–O bond. This was further supported by the Bader charge analysis. It was shown that the surface O can bind more strongly to its neighboring Co atoms (Supplementary Table 1), and therefore, it cannot be more willing to accept electrons from the adsorbing H atom, thus weakening the H adsorption (Fig. 1b). Therefore, the appropriate H* adsorption can be achieved by exerting an optimum magnitude of tensile strain on the {111}-Ov surface to fine-tune its electronic structure and achieve a high intrinsic HER activity in the CoO.

**Preparation of CoO NRs with strained and O-vacancy-rich surface.** To verify the above theoretical predictions, we attempted to attain CoO nanomaterials with strained and O-vacancy-enriched surfaces by using cation exchange methodology[48,49] where zinc oxide (ZnO) NRs acted as sacrificial templates (Supplementary Fig. 9). It is likely that during the cation exchange process, large strains and abundant vacancies were created on the surface[48]. This facile method afforded CoO NRs with an average diameter of ~100 nm on various conductive substrates (including carbon fiber paper, stainless-steel mesh, and carbon nanotube film, Supplementary Figs. 10 and 11). This offers the distinct advantage of facilitating their integration into electrochemical devices.

The complete cation exchange of ZnO NRs (Supplementary Fig. 12) is accompanied by a morphology evolution from the smooth surface of ZnO to the sawtooth-like surface of CoO, which is terminated with {111} facets (Fig. 2a and Supplementary Fig. 13). This process may introduce significant strain in the nanostructure[48,49], which most likely causes the aforementioned surface morphology evolution. As expected, an apparant curvature of {111} planes progresses from the inside to the surface of the nano-sawtooth structure (Supplementary Fig. 14). Certainly, this nanoscale bending of a crystal lattice suggests a large elastic strain in the resulting structure. We analyzed this lattice strain using geometric-phase analysis[50]. Figure 2b presents an atomic-level high-angle annular dark-field-scanning transition electron microscopy (HADDF-STEM) image showing two adjacent nano-sawtooth planes. The lattice strain components $\varepsilon_{11}$ (in {111} plane) and $\varepsilon_{22}$ (perpendicular to {111} plane) associated with the expansion/contraction of the respective lattice vectors $R_1$ and $R_2$, shown in Fig. 2b, are presented in Fig. 2c, e, and the sheer strain component $\varepsilon_{12}$ is shown in Fig. 2d. As can be seen from these figures, the values of $\varepsilon_{11}$, $\varepsilon_{12}$, and $\varepsilon_{22}$ are approximately zero in the inner part of the NR, whereas they gradually increase to large postive values on the outermost surface of the nano-sawtooth structure, indicating that the strain present on this surface is assumed to be biaxial. A detailed analysis of the strain components reveals the average values of $\varepsilon_{11} = 0.033$, $\varepsilon_{12} = 0.04$, and $\varepsilon_{22} = 0.035$ in the 2–3-nm surface region of the nano-sawtooth structure (Supplementary Fig. 15 and Supplementary Note 2). The net change in the lattice parameter of the surface of the as-synthesized CoO NRs was estimated to be ~3.0% based on the X-ray diffraction (XRD) analysis (Supplementary Fig. 16 and Supplementary Note 3), which is consistent with the observations by HAADF-STEM. Notably, the lattice strain induced by cation exchange is different from the epitaxial strain generated in the core–shell structures of the well-known dealloyed Pt-based

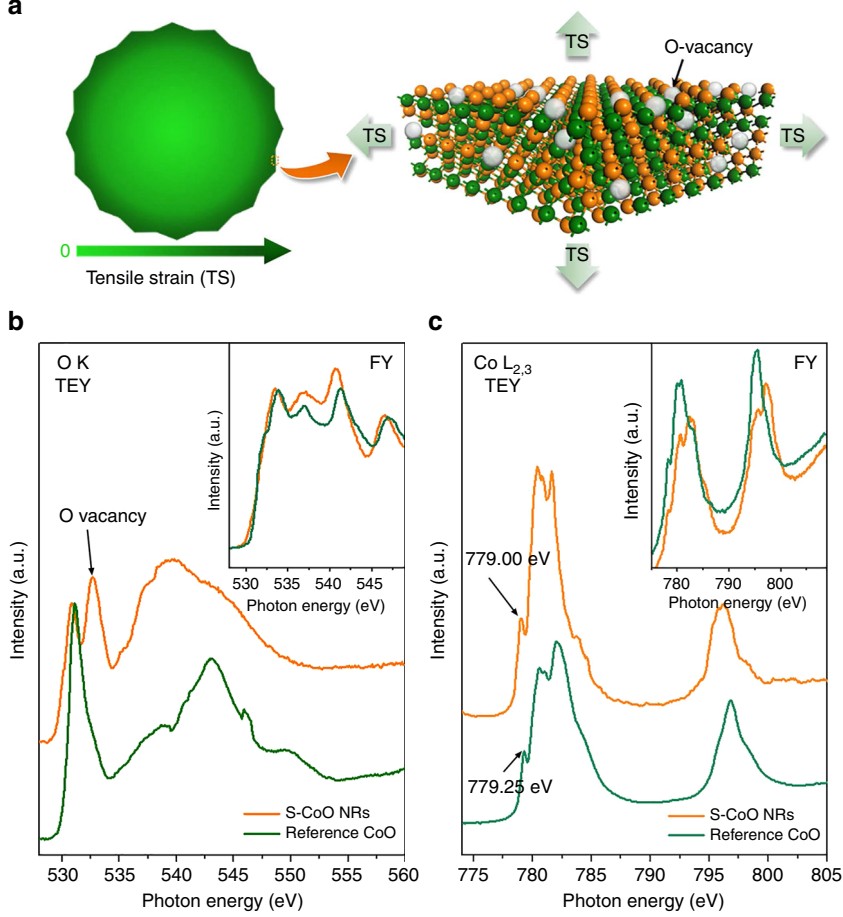

**Fig. 3** Analysis of O-vacancies on the outermost surface of S-CoO nanorods. **a** Schematic illustration of the formation of abundant O-vacancies induced by tensile strain on the outermost surface of S-CoO NRs. **b**, **c** O–K edge and Co–L$_{2,3}$ edge XANES spectra of S-CoO NRs and reference CoO in TEY mode, with the inset showing the FY signals. In **b**, the emerging new peak at ~533 eV on the spectrum of S-CoO NRs refers to O-vacancies. Correspondingly, in **c**, the peaks on the spectrum of S-CoO NRs are shifted toward low photon energy in relation to the corresponding peaks obtained for the reference CoO

nanocatalysts[27,37,51], which is likely to be gradually relieved from the core–shell interface toward the outermost atomic layer of catalysts, thereby limiting the ability of the strain-assisted manipulation of the catalytic activity[27]. In contrast, the cation exchange-induced tensile strain is mainly located on the topmost surface of CoO NRs, which is beneficial for fine-tuning the surface electronic structure and thus, in turn, their reactivity.

It is well established that the tensile strain resulting from the lattice expansion facilitates the formation of vacancies[42,52,53] and thus decreases the activation energy for the ion exchange process. Our DFT calculations indeed suggest that 3.0% tensile strain on the CoO {111}-O surface can reduce the formation energy of O-vacancies by ~40% (Supplementary Fig. 17) and thus facilitate the formation of a large quantity of O-vacancies on this surface (Fig. 3a). Experimental evidence of the presence of abundant O-vacancies on the surface of strained CoO NRs (hereafter, referred to as "S-CoO NRs") comes from synchrotron-based X-ray absorption near edge-fine structure (XANES) spectroscopy measurements. XANES data were recorded in both total electron yield (TEY) and fluorescence yield (FY) modes, with the former providing the surface-region-specific information and the latter giving bulk information. As illustrated in Fig. 3b, the FY O–K-edge spectra of the S-CoO NRs and the reference CoO are similar. In contrast, the peak attributed to the O-vacancies[53,54] emerges in the corresponding TEY spectrum of S-CoO NRs, indicating that O-vacancies are dominantly enriched on their surface (near below

2–5 nm). Accordingly, no discernible peak shift is observed in the FY signal of Co–L$_{2,3}$ edge of S-CoO NRs as compared to that of the reference CoO. However, a 0.25-eV shift toward low photon energy is visible in the TEY signal of S-CoO NRs (Fig. 3c). Using this shift, the average concentration of O-vacancies on the surface of S-CoO NRs was estimated to be about 12.5%. As predicted by the DFT calculations, such large quantity of O-vacancies on the strained CoO {111} surface will act as active sites for HER, assuring high activity of S-CoO NRs.

**Activity and durability of S-CoO NRs**. In order to reveal the relationship between the surface strain and HER catalytic activity, S-CoO NRs with 2.7%, 3.0%, and 4.0% surface strain (Supplementary Fig. 18, Supplementary Table 2, and Supplementary Note 3) were fabricated in situ on carbon fiber paper (CFP) and used directly as the working electrode for HER in 1 M KOH. For the purpose of comparison, the catalytic activities of polycrystalline CoO NRs (P-CoO NRs, Supplementary Fig. 19) and the state-of-the-art 20 wt% Pt/C catalysts (Supplementary Fig. 20) supported on CFP were also measured under the same conditions. As can be seen in Fig. 4a, b, P-CoO NRs afford low HER activity with a large Tafel slope (164 mV dec$^{-1}$), confirming that pure CoO is an inactive HER catalyst (Supplementary Fig. 21). In contrast, CoO NRs with strained surfaces exhibit improved HER activity with reduced Tafel slopes, while the 3.0% S-CoO NRs

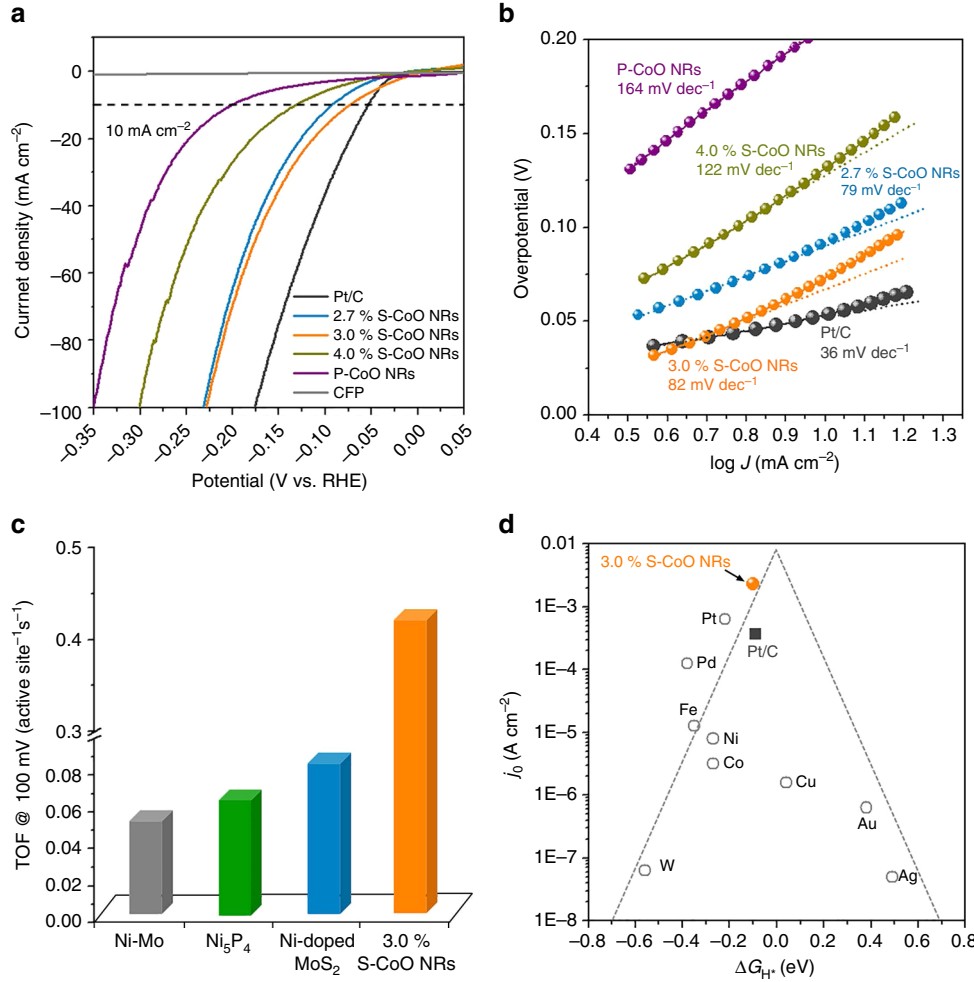

**Fig. 4** Electrocatalytic HER performance of S-CoO nanorods. **a** Linear sweep voltammetry (LSV) of S-CoO NRs with different tensile strains, P-CoO NRs, commercial Pt/C catalysts, and CFP substrate recorded in 1 M KOH solution with *iR*-correction. **b** Corresponding Tafel plots of the LSV curves in **a**. **c** Comparison of TOF values at 100-mV overpotential for Ni–Mo[59], Ni$_5$P$_4$[58], Ni-doped MoS$_2$[56], and 3.0% S-CoO NRs in alkaline solutions. **d** Volcano plots of $j_0$ measured in alkaline solution as a function of the $\Delta G_{H^*}$ for pure metals[60] (open circles) and the state-of-the-art Pt/C (closed black box), as well as the 3.0% S-CoO NRs (closed orange ball). The dashed lines are shown to guide the eye

show the highest activity (Fig. 4a, Supplementary Fig. 22, and Supplementary Note 4), consistent with the predicted optimum value of $\Delta G_{H^*} = -0.1$ eV for the 3.0% strained {111}-Ov surface (Fig. 1b and Supplementary Table 3). These observations unambiguously demonstrate that strain engineering can indeed be used to tune the activity of CoO NRs toward HER. Impressively, the optimally strained (~3.0%) S-CoO NRs display a quite small overpotential of ~ 73 mV to produce a current density of 10 mA cm$^{-2}$, comparable with that of the state-of-the-art noble metal catalysts (e.g., Pt/C, 53 mV), and even better than that of nonnoble metal-based metal alloy catalysts (e.g., Ni–Mo–N[55], 109 mV), transition metal dichalcogenides (e.g., Ni-doped MoS$_2$[56], 98 mV), and metal phosphides (e.g., CoP[57], 209 mV). A detailed performance comparison of previously reported HER catalysts demonstrates that the 3.0% S-CoO NRs are among the most active catalysts (Supplementary Table 4). Moreover, the 3.0% S-CoO NRs give ~100% Faradaic yield during the HER (Supplementary Fig. 23).

To study the intrinsic activity of the 3.0% S-CoO NRs, we estimated the turnover frequency (TOF) by normalizing the rate of H$_2$ generation to the total number of O-vacancies on CoO NRs (Supplementary Note 5). At an overpotential of 100 mV (Fig. 4c), the 3.0% S-CoO NRs exhibit an extremely high TOF of 0.41 s$^{-1}$,

which is better than the corresponding value of the well-developed benchmark HER catalysts (Ni-doped MoS$_2$[56]: 0.08 s$^{-1}$, Ni$_5$P$_4$ catalysts[58]: 0.06 s$^{-1}$, and Ni–Mo catalysts[59]: 0.05 s$^{-1}$). Notably, the true TOF of the active sites on the surface of the 3.0% S-CoO NRs would be higher, as the use of the total number of O-vacancies for TOF normalization provides a lower-bound limit. This is because it assumes that all the O-vacancies present on the surface of the S-CoO NRs are considered as active sites in the HER process.

To gain further insight into the catalytic nature of the 3.0% S-CoO NRs, we have incorporated the calculated $\Delta G_{H^*}$ (for exerted 3.0% strain on the {111}-Ov surface) along with the exchange current density, $j_0$, for the 3.0% S-CoO NRs to the volcano-shaped plot established for HER catalysts (Fig. 4d). As can be seen, the activity of the 3.0% S-CoO NRs in alkaline solutions, assessed on the basis of both $\Delta G_{H^*}$ and $j_0$, surpasses those of the common noble and nonnoble metals in alkaline solutions[60]. Strikingly, the activity of the 3.0% S-CoO NRs even exceeds that of the state-of-the-art Pt/C catalysts in alkaline solutions (Supplementary Fig. 24). Further experimental and theoretical results emphasize that the advantageous intrinsic alkaline HER activity of the 3.0% S-CoO NRs over Pt catalysts (Supplementary Fig. 25) originates from the facilitation of water dissociation on

these NRs (Supplementary Figs. 26 and 27, and Supplementary Note 6), and from the optimal $\Delta G_{H*}$ on the surface of the 3.0% S-CoO NRs as compared to the corresponding characteristics of the Pt surface (Supplementary Fig. 28).

Finally, we evaluated the long-term durability of the 3.0% S-CoO NRs. A slight HER current attenuation of ~8% was observed after 28 h of continuous testing (Supplementary Fig. 29a). The durability of the 3.0% S-CoO NRs was further confirmed by an accelerated durability test (ADT), which showed a very small negative shift of the HER polarization after 1000 continuous potential cycles (Supplementary Fig. 29b). After the ADT test, the morphology and the structure of the 3.0% S-CoO NRs are intact, as evident from scanning electron microscopy (SEM) images, XRD, and X-ray photoelectron spectroscopy results (Supplementary Figs. 30 and 31). The durability of the S-CoO NRs originates from the fact that the majority of the active sites, O-vacancies, are well preserved during the HER (Supplementary Fig. 31), and the direct growth of the CoO NRs on the CFP prevents their aggregation during the long-term reaction (Supplementary Fig. 10a–c), which is highly beneficial for the practical implementation of these materials in electrochemical devices[19,61].

## Discussion

Our study suggests that the introduction of tensile strain into the surface of CoO NRs can turn an inactive material into a highly efficient electrocatalyst toward HER. The electrocatalytic performance of this material is competitive with that of the best alkaline HER electrocatalysts reported so far. On the basis of experimental observations and theoretical calculations, we demonstrate that the tensile strain located on the outermost surface of the CoO NRs results in the creation of a large quantity of O-vacancies that facilitate water dissociation, and modulates the electronic structure to weaken hydrogen adsorption toward the optimum region. We emphasize that the activity of the strained CoO NRs is located in close proximity to the top of the volcano-shaped plot for HER catalysts; further performance enhancement can be anticipated by improving the electronic conductivity of CoO NRs. Our results illustrate the potential of tuning the surface reactivity of TMOs by controlling the strain of nanostructured TMOs that exhibit strain-driven modulations in atomic and electronic structures. These findings may open a new avenue for the development of next-generation high-performance TMO-based electrocatalysts.

## Methods

**Synthesis of electrocatalysts on various conductive substrates**. S-CoO NRs were fabricated directly on CFP using the cation exchange methodology in the gas phase[43]. In this work, CoO NRs were fabricated on both sides of CFP, and the loading of the as-synthesized S-CoO NRs was ~ 0.48 mg cm$^{-2}$. The synthetic procedure of S-CoO NRs on a carbon nanotube film and stainless-steel mesh (Supplementary Fig. 10) is the same as on CFP. P-CoO NRs were grown on CFP using a hydrothermal method as detailed elsewhere[62]. The loading of P-CoO NRs on CFP was ~ 0.46 mg cm$^{-2}$. A commercial 20 wt% Pt/C catalyst (purchased from the Fuel Cell Store), was dispersed in ethanol for at least 30 min with sonication to obtain a homogeneous ink, and ~0.4 mg of Pt/C was loaded per 1 cm$^2$ of CFP. The loading of S-CoO NRs, P-CoO NRs, and Pt/C was determined using inductively coupled plasma mass spectrometry (ICP-MS, Perkin-Elmer, NexION 300Q).

**Materials characterization**. SEM and TEM images were carried out on a Hitachi S-4800 SEM and a JOEL 2100 TEM, respectively. HAADF-STEM imaging was performed using a JEOL ARM200F microscope with the STEM aberration corrector operated at 200 kV. The convergent semiangle and collection angle were 21.5 and 200 mrad, respectively. The aberration coefficient ($C_s$) used was equal to 1 μm. Geometric-phase analysis to obtain the strain information on the surface of S-CoO NRs was conducted with Digital Micrograph software. The synchrotron-based XANES measurements were carried out using the soft X-ray spectroscopy beamline at the Canadian and Beijing Synchrotron. The S-CoO NRs for HAADF-STEM imaging and XANES measurements were obtained by cation exchange method at 600 °C.

**Electrochemical characterization**. Electrochemical measurements were performed in a three-electrode electrochemical cell using electrocatalyst-loaded CFP as the working electrode, a saturated calomel electrode as the reference electrode, and a graphite rod as the counter electrode. All potentials were calibrated with respect to reversible hydrogen electrode in the high-purity hydrogen-saturated electrolyte with a Pt plate as the working electrode (Supplementary Fig. 32). The polarization curves were recorded in 1 M KOH with a scan rate of 5 mV s$^{-1}$ and corrected for the *iR* contribution within the cell.

**Computational methods**. All DFT computations were performed using Vienna Ab initio Simulation Package (VASP). The projector augmented wave pseudopotential with the Perdew–Burke–Ernzerhof exchange-correlation functional was used in the computations. For a better description of the Co ($3d$) electrons, an effective $U$ value of 3.7 eV was applied. The relevant details and references are given in Supplementary Methods section.

**Data availability**. The data that support the findings of this study are available from the corresponding author on request.

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

## Acknowledgements

This work was supported by the National Science Fund for Excellent Young Scholars (51722103), the Natural Science Foundation of China (51571149 and 21576202), the Natural Science Foundation of Tianjin city (15JCYBJC18200), the National Basic Research Program of China (2014CB931703), and the Australian Research Council (ARC) through the Discovery Project program (DP140104062, DP160104866, and DP170104464).

## Author contributions

T.L. and S.-Z.Q. conceived the project and designed the experiments. T.L. and D.-Y.Y. performed the experiments. T.L. and H.W. carried out the TEM characterization, HADDF-STEM image simulation, and analysis. T.L., Y.J., and Z.H. conducted the DFT calculations. All authors discussed the results and commented on the manuscript.

## Additional information

**Competing interests:** The authors declare no competing financial interests.

