## [Peer Review File · Nature Communications]

Figure R1a is reprinted (adapted) with permission from (ACS Nano 2014, 8, 7948-7957). Copyright (2014) American Chemical Society. Figure R1b is reprinted by permission from Macmillan Publishers Ltd: Nature Materials (Nat. Mater. 2014, 13, 26-30), copyright (2013).

Reviewer #1 (Remarks to the Author):

In this paper, the authors report a remarkable enhancement of the alkaline HER activity of CoO nanorods fabricated by surface strain engineering. The surface strained CoO NRs exhibit excellent HER activity surpassing all the previously reported HER catalysts in alkaline solutions. Moreover, according to the theoretical and experimental results, the authors demonstrated that the existence of surface strain and abundant oxygen (O) vacancies are attributed to significantly promote the electrocatalysis toward HER. This report provides a new way for the design of superior electrocatalysts for HER. Overall, the work should be of interest for publication after addressing the following issues.

1. The CoO NRs with strained and O-vacancy-enriched surface were prepared by cation exchange using the ZnO NRs as sacrificial templates. During the cation exchange process, what factors drive the formation of O-vacancies and surface strains? Whether other metal cations, such as Ni or Cu, could also achieve corresponding TMO with strains and O-vacancies on the surface? Besides, the authors should give more detailed information for the cation exchange process in the manuscript.
2. As mentioned in the manuscript, the total numbers of O-vacancies on S-CoO NRs with different strains were determined based on ICP-MS. How to implement this process? Can ICP-MS determine the numbers of O-vacancies? The authors should give more detailed illustrations.
3. The Supplementary Figure 8 was not mentioned in the manuscript.
4. The TEM images of Pt/C and Pt NPs deposited on the CFP substrate (Supplementary Figure 21) should be provided.
5. The magnitude of tensile strain of the whole S-CoO NRs was determined by the lattice strain measured from XRD results. Hence, it is necessary to provide the original XRD patterns of S-CoO NRs exchanged at different temperatures.
6. The authors claim that the O-vacancies present on the surface of S-CoO NRs are the active sites in the HER process. For this, the excellent durability of S-CoO NRs should be closely related to the O-vacancies on the surface of S-CoO NRs, but not only ascribed to the impediment of CoO aggregation. Do the O-vacancies still exist on the S-CoO NRs after long-time HER measurements?

Reviewer #2 (Remarks to the Author):

Overall, this paper is well laid out and written. It is an interesting work, the following are some suggestions that need to be addressed before it can be published:

1. You note that the cation exchange-induced CoO nanorods have tensile strain located on the topmost surface of CoO nanorods. Why? More detailed mechanisms are needed.
2. Is there any possibility of Zn-doping on the CoO or residue of ZnO on the CoO during the cation exchange reaction? Please put EDS mapping results in Supporting Information.
3. When you give a more strain such as 4 or 5% in your calculation (Figure 1b), H adsorption is weakened? Your experimental result of 4% strained CoO showed decreased activity compared to 3%. Put the calculation results of 4 and 5% and compare with the experimental result.
4. Although I believe your stability test and XRD result in Figure 23 and 24 in Supporting Information, why does your CoO not reduce to Co during the electrochemical reaction at the more negative potential (-0.6V)? The Co^{2+} is electrochemically reduced to Co at -0.28V (vs. RHE).
5. How did you change to RHE from Hg/HgO reference electrode? Put the calculation in Supporting Information.
6. Please put X-ray Rietveld refinement results in Figure 14 (Supporting Information) for the change in the lattice parameters.

7. There are no notifications about Figure S8. Please put in Page 5.

Reviewer #3 (Remarks to the Author):

This manuscript reports an interesting discovery that tensile strain turns CoO into an active HER catalyst as Pt. The conclusion is supported by complimentary theoretical and experimental results. Therefore, the manuscript is recommended for publication after addressing the following questions.

1. The effect of tensile strain on $\Delta G(H)$ in Fig. 1b is based on pristine CoO. However, in Fig. 3, it discusses that strain leads to generation of O-vacancy that serves as the active site. Then the $\Delta G(H)$ on those O-vacancy sites needs to be provided as a function of strain and amount of O-vacancy.
2. The faraday efficiency of HER needs to be measured to confirm that there are no other side reactions going on.
3. The introduction needs to explain why this work chooses CoO, not other oxides, to test the strain effect.

Response to Reviewer #1

General Comments:

In this paper, the authors report a remarkable enhancement of the alkaline HER activity of CoO nanorods fabricated by surface strain engineering. The surface strained CoO NRs exhibit excellent HER activity surpassing all the previously reported HER catalysts in alkaline solutions. Moreover, according to the theoretical and experimental results, the authors demonstrated that the existence of surface strain and abundant oxygen (O) vacancies are attributed to significantly promote the electrocatalysis toward HER. This report provides a new way for the design of superior electrocatalysts for HER. Overall, the work should be of interest for publication after addressing the following issues.

Response:

We would like to thank the Reviewer for his/her valuable comments and positive recommendation.

Original comment 1-1:

The CoO NRs with strained and O-vacancy-enriched surface were prepared by cation exchange using the ZnO NRs as sacrificial templates. During the cation exchange process, what factors drive the formation of O-vacancies and surface strains? Whether other metal cations, such as Ni or Cu, could also achieve corresponding TMO with strains and O-vacancies on the surface? Besides, the authors should give more detailed information for the cation exchange process in the manuscript.

Response:

Overall, the cation exchange drives the formation of O-vacancies and surface strains on S-CoO NRs, which in turn facilitate the cation exchange. The creation of vacancies during ion exchange process has been well investigated, for instance, the exchange of Pb for Cd on the surface of PbS (*ACS Nano* 2014, 8, 7948-7957). As seen in **Fig. R1a**, the vacancies control the communication between the parent PbS and product CdS crystals. On the other hand, it has been reported that the surface strain on the Fe nanocrystals can significantly lower the activation energies for the diffusion of ions (**Fig. R1b**, *Nat. Mater.* 2014, 13, 26-30). Therefore, the presence of surface strain on CoO NRs facilitated the Co^{2+} inward diffusion and Zn^{2+} outward diffusion through O-vacancies (**Fig. R2**).

Figure R1. **a**, Schematic representation of the exchange of Pb for Cd on the surface of PbS quantum dots (*ACS Nano* 2014, 8, 7948-7957). **b**, Schematic illustration of surface strain facilitating the O^{2-} inward diffusion and Fe^{2+} outward diffusion (*Nat. Mater.* 2014, 13, 26-30).

Figure R2. Schematic illustration of creation of O-vacancies and strain on the surface of CoO by cation exchange methodology.

Theoretically, in order to well preserve the framework of the parent crystal during the ion exchange process, the parent and product crystals should have both similar crystal structures and lattice parameters. In this point of view, NiO with surface strain and O-vacancies may be accessible, since the parent ZnO and product NiO crystals both are close-packed structures, and the ionic radii of Ni^{2+} and Co^{2+} are similar in tetrahedral coordination (0.74 Å and 0.72 Å, respectively) and in octahedral coordination (0.88 Å and 0.89 Å, respectively). For CuO, since the zinc blend structure is much different from that of ZnO (hexagonal structure), it is maybe difficult to achieve CuO through cation exchange method using ZnO as a sacrificial template.

Following the comments of the Reviewer, we have added the above **Fig. R2** as **Supplementary Fig. 9** in the revised supporting information to give more description and better illustrate the cation exchange method and its role in the creation of O-vacancies and strain on the surface of CoO.

Original comment 1-2:

As mention in manuscript, the total numbers of O-vacancies on S-CoO NRs with different strains were determined based on ICP-MS. How to implement this process? Can the ICP-MS determine the numbers of O-vacancies? The authors should give more detail illustrations.

Response:

After the cation exchange process, the as-exchanged CoO NRs were immediately taken out of the tube furnace and the total mass of CoO NRs (m_{CoO}) was accurately determined using an analytical balance. Then, the resulting CoO NRs were totally dissolved in HNO₃, and the mass of Co²⁺ ions ($m_{\text{Co}^{2+}}$) was determined by ICP-MS. The O-vacancy concentration (δ) of CoO NRs can be estimated by the following equation,

$$\delta = 1 - \frac{(m_{\text{CoO}} - m_{\text{Co}^{2+}}) / M_{\text{O}}}{m_{\text{Co}^{2+}} / M_{\text{Co}}}$$

where M_{Co} and M_{O} are the molecular weights of Co and O, respectively.

Accordingly, we have added above description of the determination of O-vacancies by ICP-MS in the **Supplementary Methods** in the revised supporting information.

Original comment 1-3:

The Supplementary Figure 8 was not mentioned in manuscript.

Response:

We thank the Reviewer for pointing out this issue. We have added the citation of Supplementary Fig. 8 in the revised manuscript **on page 5, line 11**.

‘As can be seen, the tensile strain upshifts the O 2p-band of CoO (Supplementary Fig. 8), which results in greater covalency of the Co-O bond.’

Original comment 1-4:

The TEM images of Pt/C and Pt NPs deposited on the CFP substrate (Supplementary Figure 21)

should be provided.

Response:

According to the suggestion of the Reviewer, the Pt/C and Pt NPs deposited on CFP substrates were characterized by SEM and TEM. As seen in **Fig. R3** and **Fig. R4**, Pt/C and Pt NPs are uniformly distributed on CFP, respectively.

Figure R3. Characterization of Pt/C catalysts supported on CFP substrate. **a**, SEM image. **b**, HRTEM image.

Figure R4. Characterization of Pt catalysts supported on CFP substrate. **a**, SEM image. **b**, HRTEM image.

Accordingly, we have added Figs. R3 and R4 as **Supplementary Figs. 20 and 25**, respectively, in the revised supporting information.

Original comment 1-5:

The magnitude of tensile strain of the whole S-CoO NRs was determined by the lattice strain measured from XRD results. Hence, it is necessary to provide the original XRD patterns of S-CoO NRs exchanged at different temperature.

Response:

Following the suggestion of the Reviewer, we have added the XRD patterns of S-CoO NRs exchanged at different temperatures and P-CoO NRs (**Fig. R5**) as **Supplementary Fig. 18** in the revised supporting information.

Figure R5. XRD spectra of S-CoO NRs prepared by cation exchange method at different temperatures and P-CoO NRs.

Original comment 1-6:

The authors claim that the O-vacancies present on the surface of S-CoO NRs are the active sites in the HER process. For this, the excellent durability of S-CoO NRs should be closely related to the O-vacancies on the surface of S-CoO NRs, but not only ascribed to the impediment of CoO aggregation. Do the O-vacancies still exist on the S-CoO NRs after long-time HER measurements?

Response:

We agree with the Reviewer that the excellent durability of S-CoO NRs should be ascribed to both the stability of O-vacancies and impediment of CoO aggregation. XPS analysis was conducted to discern the stability of O-vacancies during HER. It is well known that O-vacancies can donate electrons to adjacent metal ions, causing peak shifts in XPS, EELS or XANES spectra of metal (*Nature*, 2004, 430, 657-661; *Adv. Funct. Mater.* 2016, 26, 1564-1570). As can be seen in **Fig. R6** (original Supplementary Fig. 25), no noticeable peak shift is observed on the Co 2p XPS spectrum of 3.0 % S-CoO NRs after stability test as compared to that of the fresh sample. Therefore, we concluded that the majority of O-vacancies are preserved during HER.

Figure R6. XPS Co 2p spectra of 3.0 % S-CoO NRs before and after ADT test.

Following the Reviewer's comment, we have added the above discussion about the presence of O-vacancies after stability test in the revised manuscript **on page 9, line 24 to page 10, line 1**,

'This excellent durability of S-CoO NRs originates from the fact that majority of active sites, O-vacancies, are well preserved in HER (Supplementary Fig.31), and the direct growth of CoO NRs on CFP to prevent their aggregation during long-term reaction (Supplementary Fig. 10a-10c), which is highly beneficial for practical implementation of these materials in electrochemical devices^{19,56}.

and in the **legend** of original Supplementary Fig. 25 (**Supplementary Fig. 31** in the revised supporting information).

‘Moreover, no noticeable peak shift was observed on the Co 2p XPS spectrum of 3.0 % S-CoO NRs after stability test in comparison to that of the fresh sample, suggesting that the majority of O-vacancies are preserved during the HER.’

Response to Reviewer #2

General Comments:

Overall, this paper is well laid out and written. It is an interesting work, the following are some suggestions need to be addressed before it can be published.

Response:

We would like to thank the Reviewer for his/her valuable comments to help us to improve the quality of this manuscript.

Original comment 2-1:

You note that the cation exchange-induced CoO nanorods have tensile strain located on the topmost surface of CoO nanorods. Why? More detailed mechanisms are needed.

Response:

It has been reported that the surface tensile strain on the Fe nanocrystals can significantly lower the activation energies for ion diffusions (*Nat. Mater.* 2014, 13, 26-30). A similar situation is in the system studied; namely, the presence of surface strain on CoO NRs facilitates the Co^{2+} inward diffusion and Zn^{2+} outward diffusion through O-vacancies (**Fig. R7**), which results in fast cation exchange. Therefore, we concluded that the cation exchange drives the formation of tensile strain on the topmost surface of CoO NRs.

Following the suggestion of the Reviewer, we have added Fig. R7 as **Supplementary Fig. 9** in the revised supporting information to improve the description of cation exchange method and its role in the creation of surface strain on CoO NRs.

Figure R7. Schematic illustration of creation of O-vacancies and strain on the surface of CoO by cation exchange methodology. The presence of surface strain on CoO NRs facilitates the Co^{2+} inward diffusion and Zn^{2+} outward diffusion through O-vacancies, which results in the fast cation exchange.

Figure R8. Characterization of 3.0 % S-CoO NRs. **a**, EDS mapping of an individual 3.0 % S-CoO NR. It shows that ZnO has been completely transformed into CoO. **b** and **c**, XPS and XRD patterns of 3.0 % S-CoO NRs, respectively.

Original comment 2-2:

Is there any possibility of Zn-doping on the CoO or residue of ZnO on the CoO during cation exchange reaction? Please put EDS mapping result in Supporting Information.

Response:

Our EDS mapping and XPS spectrum of 3.0 % S-CoO NRs with the best HER performance show that only Co and O elements are in the as-exchanged CoO NRs (**Figs. R8a and R8b**). Further XRD characterization confirms that no second phase exists in CoO NRs (**Fig. R8c**). These collective results demonstrate that ZnO template was completely transformed into CoO, and there is no residue of ZnO in the as-exchanged 3.0 % S-CoO NRs.

According to the comment of the Reviewer, we have added Fig. R8 as **Supplementary Fig. 12** in the revised supporting information.

Original comment 2-3:

When you give a more strain such as 4 or 5 % in your calculation (Figure 1b), H adsorption is weakened? Your experimental result of 4% strained CoO showed decreased activity compared to 3%. Put the calculation results of 4 and 5% and compare with experimental result.

Response:

According to the suggestion of the Reviewer, we have added the H adsorption energy of 4 % strained CoO in **Fig. R9a**. As can be seen, H* adsorption is continuously weakened with increasing magnitude of the applied strain, with 3 % strained CoO exhibiting the optimum ΔG_{H^*} of -0.10 eV. Accordingly, 3.0 % S-CoO NRs show the best HER performance (**Fig. R9b**). Therefore, the theory and experiment are in an excellent agreement.

Following the suggestion of the Reviewer, we have replaced original **Fig. 1b** with Fig. R9a in the revised manuscript. The related comparison of the calculated hydrogen adsorption free energy and HER performance is listed on **page 8, lines 8-12**.

‘In contrast, CoO NRs with strained surfaces exhibit advantageous HER activity with reduced Tafel slopes, while 3.0 % S-CoO NRs show the highest activity (Fig. 4a, Supplementary Fig. 22 and Supplementary Note 4), consistent with the predicted optimum value of $\Delta G_{H^} = -0.1$ eV for 3.0 % strained {111}-Ov surface (Fig. 1b and Supplementary Table 3).’*

Figure R9. **a**, Hydrogen adsorption free energy, ΔG_{H^*} , versus tensile strain for the CoO {111}-Ov surface. **b**, Linear sweep voltammetry (LSV) of S-CoO NRs with different tensile strains.

Original comment 2-4:

Although I believe your stability test and XRD result in Figure 23 and 24 in Supporting Information, why your CoO doesn't reduce to Co during electrochemical reaction at the more negative potential (-0.6 V)? The Co^{2+} is electrochemically reduced to Co at -0.28V (vs. RHE).

Response:

We think that the reason of a good stability of S-CoO NRs during current potential range is closely related to their specific atomic structure. As can be seen in **Fig. 2a**, the topmost surface of S-CoO NRs is enclosed with {111} nanofacets, which are terminated with pure O atomic planes (*Nat. Commun.* 2016, 7, 12876). Hence, the chemical nature of as-synthesized CoO is different from traditional CoO as mentioned by the Reviewer.

Moreover, to further confirm the stability of our S-CoO NRs in the potential range of -0.8-0 V_{RHE} , cyclic voltammetry (CV) curve was measured. As seen in **Fig. R10**, there is no visible peaks observed and attributed to reduction of Co^{2+} . Furthermore, we calculated the Faradaic efficiency of 3.0 % S-CoO NRs at -0.18 and -0.6 V_{RHE} by measuring the H_2 evolved from the HER cell. As can be seen in **Fig. R11**, 3.0 % S-CoO NRs give ~100% Faradaic yield at both -0.18 and -0.6 V_{RHE} , indicating no other side reactions going on. Therefore, the collective results demonstrate the excellent stability of S-CoO NRs during HER.

Figure R10. Cyclic voltammetry (CV) curve of 3.0 % S-CoO NRs between -0.8-0 V_{RHE} with a scan rate of 10 mV s⁻¹.

Figure R11. Faradaic efficiency of 3.0 % S-CoO NRs from gas chromatography measurement of evolved H₂.

According to the comment of the Reviewer, we have added **Fig. R11** as **Supplementary Fig. 23** in the revised supporting information, and a new sentence in the main text to describe the Faradaic efficiency of 3.0 % S-CoO NRs in the revised manuscript **on page 8, lines 20-21**.

‘Moreover, 3.0 % S-CoO NRs give ~100% Faradaic yield during HER (Supplementary Fig. 23).’

Moreover, the experimental details of characterization the Faradaic efficiency have been added in the **Supplementary Methods** in the revised supporting information.

‘The Faradaic yield was calculated from the total charge $Q(C)$ passed through the cell and the total

amount of hydrogen produced n_{H_2} (mol). The total amount of hydrogen produced was measured using gas chromatography (Agilent 7890B). Assuming that two electrons are used to produce one H_2 molecule, the Faradaic efficiency can be calculated as follows:

$$\eta = \frac{2F \times n_{H_2}}{Q}$$

where F is the Faraday constant.'

Original comment 2-5:

How did you change to RHE from Hg/HgO reference electrode? Put the calculation in Supporting Information.

Figure R12. Calibration of the reference saturated calomel electrode (SCE).

Response:

Saturated calomel electrode (SCE) was used as the reference electrode in all measurements. The calibration of SCE respect to reversible hydrogen electrode (RHE) was conducted according to the literature (*Nat. Mater.* 2011, 10, 780-786). Specifically, the calibration was performed in hydrogen saturated electrolyte with a Pt sheet as the working electrode. Cyclic voltammetry run at a scan rate of 1 mV s^{-1} (**Fig. R12**), and the average of the two potentials at which the current value was zero was taken as the thermodynamic potential. Therefore, in 1 M KOH, $V_{RHE} = V_{SCE} + 1.03 \text{ V}$.

Following the Reviewer's comment, we have added **Fig. R12** as **Supplementary Fig. 32** and the above description of calibration of SCE in **the legend of Supplementary Fig. 32** in the revised supporting information.

Original comment 2-6:

Please put X-ray Rietveld refinement result in Figure 14 (Supporting Information) for the change in the lattice parameters.

Response:

Actually, the lattice parameters of S-CoO NRs exchanged at different temperatures and P-CoO in original Supplementary Table 2 were X-ray Rietveld results. We have corrected the related note in **Supplementary Table 2** in the revised supporting information.

Original comment 2-7:

There are no notifications about Figure S8. Please put in Page 5.

Response:

We thank the Reviewer for pointing this issue. We have added the citation of Supplementary **Fig. 8** in the revised manuscript **on page 5, line 11**.

'As can be seen, the tensile strain upshifts the O 2p-band of CoO (**Supplementary Fig. 8**), which results in greater covalency of the Co-O bond.'

Response to Reviewer #3

General Comments:

This manuscript reports an interesting discovery that tensile strain turns CoO into an active HER catalyst as Pt. The conclusion is supported by complimentary theoretical and experimental results. Therefore, the manuscript is recommended for publication after addressing the following questions.

Response:

We would like to thank the Reviewer for his/her helpful comments and positive recommendation.

Original comment 3-1:

The effect of tensile strain on $\Delta G(H)$ in Fig. 1b is based on pristine CoO. However, in Fig. 3, it discusses that strain leads to generation of O-vacancy that serves as the active site. Then the $\Delta G(H)$ on those O-vacancy sites needs to be provided as a function of strain and amount of O-vacancy.

Response:

The effect of tensile strain on ΔG_{H^*} in **Fig. 1b** is based on CoO with ~11.1 % surface O-vacancies (described in original main text **on page 5, line 3** and **in legend of Fig. 1b**). When building the computational super cell, we kept the concentration of surface O-vacancies consistent with the experimental data (~12.5 %). The calculated ΔG_{H^*} and experimental HER performance are in good agreement with each other that 3.0 % strained CoO exhibits the optimal $\Delta G_{H^*}=-0.1$ eV and 3.0 % S-CoO NRs afford the best HER performance. Therefore, the current structural model matches the real catalyst, and our DFT study provides guidance for interpretation of the results.

Accordingly, to address this comment, we have added a sentence to describe the consideration of amounts of O-vacancies when building the computational super cell in **Supplement Methods** in revised supporting information.

‘For this unit cell size, the surface O-vacancy concentration is 11.1% (defined as the number of S-vacancies divided by the total number of O atoms on the pristine surface), which is in line with the experimental data (~12.5 %) estimated based on the Co-L_{2,3} edge XANES spectra (Fig. 3c).’

Original comment 3-2:

The faraday efficiency of HER needs to be measured to confirm that there are no other side reactions going on.

Response:

Faradaic efficiency of 3.0 % S-CoO NRs was calculated by measuring the H₂ evolved from the HER cell. As shown in **Fig. R13**, 3.0 % S-CoO NRs give ~100% Faradaic yield with both relative low (28 mA cm⁻²) and high (~210 mA cm⁻²) HER current, indicating no other side reactions going on during HER.

Figure R13. Faradaic efficiency of 3.0 % S-CoO NRs from gas chromatography measurement of evolved H₂.

Following the Reviewer’s comment, we have added **Fig. R13** as **Supplementary Fig. 23** in the revised supporting information, and a new sentence in the main text to describe the Faradaic efficiency of 3.0 % S-CoO NRs in the revised manuscript **on page 8, lines 20-21**.

‘Moreover, 3.0 % S-CoO NRs give ~100% Faradaic yield during HER (Supplementary Fig. 23).’

Moreover, the experimental details of characterization the Faradaic efficiency were added in the **Supplementary Methods** in the revised supporting information.

‘The Faradaic yield was calculated from the total charge $Q(C)$ passed through the cell and the total amount of hydrogen produced n_{H_2} (mol). The total amount of hydrogen produced was measured using gas chromatography (Agilent 7890B). Assuming that two electrons are used to produce one H₂ molecule, the Faradaic efficiency can be calculated as follows:

$$\eta = \frac{2F \times n_{H_2}}{Q}$$

where F is the Faraday constant.’

Original comment 3-3:

The introduction needs to explain why this work chooses CoO, not other oxides, to test the strain effect.

Response:

Following the suggestion of the Reviewer, we have added a new sentence in the Introduction

part to introduce the research background of CoO in the revised manuscript **on page 3, lines 19-20**.

‘As an emerging catalytic material for OER⁴⁰ and photocatalytic hydrogen evolution⁴¹, cobalt (II) oxide (CoO) has received considerable attention.’

Moreover, two new references have been added in the main text and in the Reference list.

END OF RESPONSE

Reviewer #1 (Remarks to the Author):

The authors have carefully addressed all my concerns, therefore this manuscript can be readily for acceptance.

Reviewer #2 (Remarks to the Author):

They made improved revision. And, this paper should be of interest after addressing the following minor issues.

1. The strained CoO NRs were prepared by cation exchange using the ZnO nanorods. During the cation exchange process (Supplementary Fig.9), the ZnO and CoO have similar structure. But that explanation is somewhat weird. The ZnO has hexagonal close packed structure (hcp) but CoO has cubic close packed structure (fcc) as a stable structure. It is also well matched in your XRD. So phase transition should be happen during the cation exchange. You had better redesign the cation exchange processes, and the formation mechanism of tensile strain on the topmost surface of CoO NRS.

2. There are several papers about strain effect for water splitting (water oxidation or proton reduction) based on transition metal oxides. You should introduce some related strain effect based on transition metal oxides such as Fe and Mn.

3. In Page 5, there are several "Ov" such as {111}-Ov. Is it "oxygen vacancy" or "11.1% surface O-vacancies"?

Reviewer #3 (Remarks to the Author):

The authors have addressed all the comments and the manuscript is recommended for publication.

Response to Reviewer #2

General Comments:

They made improved revision. And, this paper should be of interest after addressing the following minor issues.

Response:

We would like to thank the Reviewer for his/her helpful comments and positive recommendation.

Original comment 1:

The strained CoO NRs were prepared by cation exchange using the ZnO nanorods. During the cation exchange process (Supplementary Fig.9), the ZnO and CoO have similar structure. But that explanation is somewhat weird. The ZnO has hexagonal close packed structure (hcp) but CoO has cubic close packed structure (fcc) as a stable structure. It is also well matched in your XRD. So phase transition should be happened during the cation exchange. You had better redesign the cation exchange processes, and the formation mechanism of tensile strain on the topmost surface of CoO NRs.

Response:

Actually, we have characterized the crystal structures of products at different exchanged stages by XRD (Fig. R1), and phase transition indeed happened during the cation exchange process as mentioned by the Reviewer. It is found that at the initial stage of cation exchange, hexagonal ZnO and fcc CoO coexist in the exchanged products (dark yellow and purple curves). When the molar proportion of Zn^{2+} in the resultant product is less than 10% (measured using inductively coupled plasma mass spectrometry), only fcc structure without visible defects, second phase, or precipitation is observed in the exchanged product (orange curve). Therefore, the residual Zn^{2+} ions are well-integrated into the fcc lattice of CoO to form solid solution $Zn_xCo_{1-x}O$. Reasonably, the ionic radii of Zn^{2+} and Co^{2+} are similar in octahedral coordination (0.88 Å and 0.89 Å, respectively) (*Chem. Mater.* 2012, 24, 2311-2315). Hence, in original Supplementary Fig. 9, the ZnO and CoO have the similar structure.

Regarding the formation mechanism of tensile strain, it has been reported that the surface strain on the Fe nanocrystals can significantly lower the activation energies for diffusion of ion (*Nat. Mater.* 2014, 13, 26-30). A similar situation is systematically studied here. The cation exchange drives the formation of O-vacancies and surface strains on S-CoO NRs, which in turn facilitate the cation exchange.

Figure R1. XRD patterns of products at different exchanged stages.

Figure R2. Schematic illustration of creation of O-vacancies and strain on the surface of CoO (viewed from [100] direction) by cation exchange methodology. The presence of surface strain on CoO NRs facilitates the Co^{2+} inward diffusion and Zn^{2+} outward diffusion through O-vacancies, which results in the fast cation exchange. Notably, at the initial stage of cation exchange, hexagonal ZnO and face-centered cubic (fcc) CoO coexist in the exchanged product. As replacement reaction of the Co^{2+} cations for Zn^{2+} cations proceeds, phase transition happens, and the residual Zn^{2+} ions are well-integrated into the fcc lattice of CoO to form solid solution $\text{Zn}_x\text{Co}_{1-x}\text{O}$.

We have redesigned the schematic illustration of cation exchanged process, in which the molar composition ratio of $\text{Zn}^{2+}/\text{Co}^{2+}$ in exchanging $\text{Zn}_x\text{Co}_{1-x}\text{O}$ (fcc structure) is less than 10%. Following the suggestion of the Reviewer, we have renewed **Supplementary Fig. 9** as Fig. R2 shown here.

Moreover, the discussion about the phase transition during cation exchange process was added in **the caption of Supplementary Fig. 9** in the revised supporting information.

‘Notably, at the initial stage of cation exchange, hexagonal ZnO and face-centered cubic (fcc) CoO coexist in the exchanged product. As replacement reaction of the Co^{2+} cations for Zn^{2+} cations proceeds, phase transition happens, and the residual Zn^{2+} ions are well-integrated into the fcc lattice of CoO to form solid solution $\text{Zn}_x\text{Co}_{1-x}\text{O}$.’

Original comment 2:

There are several papers about strain effect for water splitting (water oxidation or proton reduction) based on transition metal oxides. You should introduce some related strain effect based on transition metal oxides such as Fe and Mn.

Response:

Following the Reviewer’s suggestion, we have added the introduction of strain effect based on transition metal oxides in the revised manuscript **on page 3, lines 11-12**.

‘In the case of TMOs, strain effect was introduced to modulate the electronic structure of TMOs, e.g., iron oxides³⁸⁻⁴⁰.’

Moreover, the above references were added in the revised manuscript and in the reference list.

Original comment 3:

In Page 5, there are several "Ov" such as {111}-Ov. Is it "oxygen vacancy" or "11.1% surface O-vacancies"?

Response:

{111}-Ov is O-vacancy rich {111}-O surface (with 11.1% surface O-vacancies). We have added the corresponding description in the revised manuscript **on page 5, lines 4-5**.

*‘Surprisingly, when tensile strain was exerted on **O-vacancy rich {111}-O surface (hereafter, referred to as ‘{111}-Ov surface’, with 11.1% surface O-vacancies, Supplementary Fig. 7), H^* adsorption was continuously weakened with increasing magnitude of the applied strain (Fig. 1b).**’*

END OF RESPONSE

Reviewer #2 (Remarks to the Author):

The authors have addressed all my concerns. I recommend this article for publication.

Response to the Reviewer #2

Reviewer Letter:

The manuscript has been greatly improved. All my previous concerns have been properly addressed. Therefore, I recommend this manuscript for publication in Nature Communications in its current form.

Response:

We thank the Reviewer for his/her positive comments and recommendation.